# Expanding the repertoire of human tandem repeat RNA-binding proteins

**Agustín Ormazábal[1,2,3], Matías Sebastián Carletti[2,3], Tadeo Enrique Saldaño[2,3,4], Martín Gonzalez Buitron[2,3], Julia Marchetti[2], Nicolas Palopoli[2,3], Alex Bateman[1]***

**1** European Molecular Biology Laboratory, European Bioinformatics Institute (EMBL-EBI), Wellcome Genome Campus, Hinxton, United Kingdom, **2** Departamento de Ciencia y Tecnología, Universidad Nacional de Quilmes, Bernal, Buenos Aires, Argentina, **3** Consejo Nacional de Investigaciones Científicas y Técnicas, Godoy Cruz, Buenos Aires, Argentina, **4** Departamento de Ciencias Básicas, Facultad de Agronomía, Universidad Nacional del Centro de la Provincia de Buenos Aires, Azul, Buenos Aires, Argentina

* agb@ebi.ac.uk

**Data Availability Statement:** All relevant data are within the paper and its Supporting Information files.

## Abstract

Protein regions consisting of arrays of tandem repeats are known to bind other molecular partners, including nucleic acid molecules. Although the interactions between repeat proteins and DNA are already widely explored, studies characterising tandem repeat RNA-binding proteins are lacking. We performed a large-scale analysis of human proteins devoted to expanding the knowledge about tandem repeat proteins experimentally reported as RNA-binding molecules. This work is timely because of the release of a full set of accurate structural models for the human proteome amenable to repeat detection using structural methods. The main goal of our analysis was to build a comprehensive set of human RNA-binding proteins that contain repeats at the sequence or structure level. Our results showed that the combination of sequence and structural methods finds significantly more tandem repeat proteins than either method alone. We identified 219 tandem repeat proteins that bind RNA molecules and characterised the overlap between repeat regions and RNA-binding regions as a first step towards assessing their functional relationship. We observed differences in the characteristics of repeat regions predicted by sequence-based or structure-based methods in terms of their sequence composition, their functions and their protein domains.

## Introduction

Protein regions consisting of arrays of tandem repeats are known to bind other molecular partners, including nucleic acid molecules [1]. Although the interactions between repeat proteins and DNA are already widely explored, studies intended to characterise tandem repeat RNA-binding proteins are lacking. It has been previously reported that the RNA and DNA binding proteins are enriched in proteins containing tandem repeats [2]. More recently, many tandem repeats in proteins were associated with Pfam families such as those involved in transcription, RNA-splicing or stabilisation, and multiprotein complexes assembly [2, 3]. Some examples are Zn-fingers, WD-40 repeats, the KRAB box and the KH domains, among others. The unique

**Funding:** This project has received funding from the European Union's Horizon 2020 research and innovation staff exchange programme REFRACT under grant agreement No 823886. A.O., M.S.C. and M.G.B. are Ph.D. fellows, J.M. is a postdoctoral researcher, and N.P. is an adjunct researcher from Consejo Nacional de Investigaciones Científicas y Técnicas (CONICET). The work was in part supported by funding from Agencia Nacional de Promoción Científica y Tecnológica (ANPCyT) Grant #PICT-2020-SERIEA-00192 to N.P. The authors of this work are also supported by the core EMBL funding and declare that they have no competing interests. The funders had no role in study design, data collection and analysis, decision to publish, or preparation of the manuscript. There was no additional external funding received for this study.

**Competing interests:** The authors have declared that no competing interests exist.

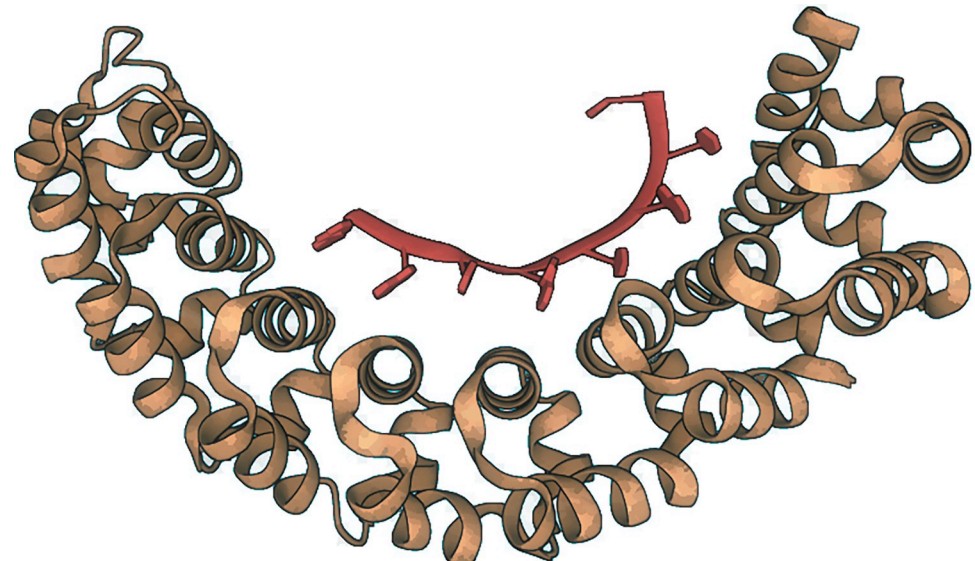

**Fig 1. The Pumilio protein (brown) forming a complex with a nanos response element RNA (red) (PDB access code: 1M8W).**

arrangements of protein repeats would provide an elegant and scalable solution to bind, read and process RNA sequences effectively, where each repeat could bind to one or a few ribonucleotides. A well-known example of an RNA-binding protein repeat is the Pumilio family of proteins. The Pumilio family of proteins contains PUF domains generally composed of eight 36-amino-acid repeats, which each bind a single ribonucleotide base. The interaction with the RNA molecule is base-specific [4], and it is established via hydrogen bonding or van der Waals contacts between the Watson-Crick edge from the nucleic acid and the amino acids at positions 12 and 16 of the PUF repeat (Fig 1) [5–9].

Among the functions upregulated by Pumilio the most widespread are embryogenesis [10], cell differentiation [11, 12], degradation of target mRNAs [13], and cytoplasmic sensing of viral infection [5], among others. The many biological functions associated with the Pumilio family are facilitated by its ability to bind the 3'-UTR of mRNA targets, thus controlling their stability and translation [8, 14]. Zinc fingers also provide a further example of small repeated domains which bind to three DNA nucleotides per zinc finger.

Several studies devoted to understanding the interactions between repeat proteins and DNA molecules were previously reported [15, 16]. It has also been previously reported that a high proportion of the human RNA-binding proteins lack a native 3D structure [17]. Previous works also established that protein-RNA interactions can induce co-folding of both molecules and structure stabilisation [18, 19]. For example, it was reported that the RGG peptide recognizes duplex-quadruplex junctions by structural complementary [19]. However, a large-scale analysis of proteins that interact with RNA and are predicted to have tandem repeats has not been carried out.

Due to the availability of high-quality structure predictions for nearly all human proteins we believe that such a comprehensive census of RNA-binding repeat proteins is now timely. With the aim to expand the knowledge about the RNA-binding repeat proteins, here we identify and characterise 219 human proteins predicted as repetitive by sequence or structure based methods, that were previously reported as RNA-binding proteins. We also explored the differences between the characteristics of the proteins predicted by sequence methods and

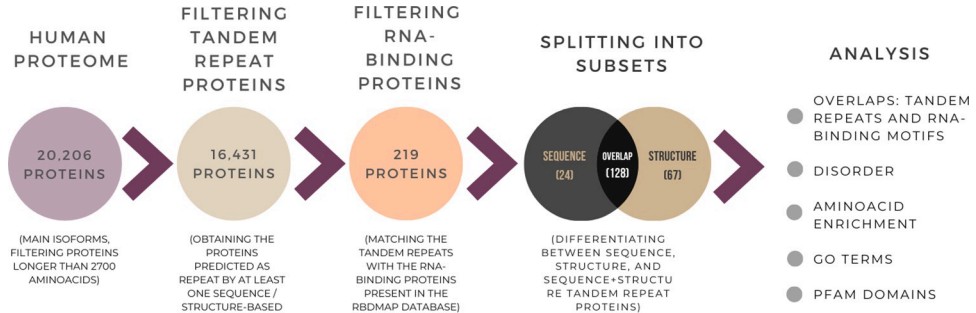

**Fig 2. Diagram showing the pipeline for finding RNA-binding repeat proteins and the analysis performed on them.**

those predicted by structure-based methods, and the proteins predicted as repetitive by both of them.

## Methods

The pipeline for finding RNA-binding repeat proteins is summarised in Fig 2.

Our study used a comprehensive dataset of human protein sequences and AlphaFold structural models [20, 21]. It consists of 20,206 protein sequences and their corresponding structures, with proteins longer than 2,700 amino acids filtered out since they do not have a full length AlphaFold model. We also did not include alternative isoforms for proteins, which are not currently available as AlphaFold models.

Prediction of tandem repeats in each protein was carried out with the computational methods TAPO [22], RepeatsDB-Lite [23], CE-Symm 2.2.2 [24], REP2 [25], TRDistiller and TRDistiller(HHrepID) [26], and TRAL (both for short tandem repeats, STRs, and for Armadillo repeats, ARM) [27]. In the following table we summarise the types of repeats detected, the extent of the repeat prediction (either single repeat or multiple repeat), and the criteria adopted by each method. We will refer to a single repeat as the smallest structural or sequence building block forming a repeat region. Here, repeat regions correspond to one or more single repeats units (Table 1).

The subset of candidate RNA-binding repeat proteins was obtained by filtering the original dataset in order to obtain those proteins predicted as containing repeats by at least one of the

**Table 1. Table summarising the repeat prediction methods used in this study.**

| Prediction tool | Repeats (Single/ Multiple) | Type of repeats predicted | Sequence/Structure based |
|---|---|---|---|
| TAPO | Single | All | Structure |
| RepeatsDB-Lite | Single | All | Structure |
| CE-Symm 2.2.2 | Multiple | All (repeats with both open and closed symmetry) | Structure |
| REP2 | Single | Ankyrin (ANK), Armadillo (ARM), HAT, HEAT, KELCH, LRR, PFTA, PFTB, RCC1, TPR and WD40 | Sequence |
| TRDistiller | Multiple | All | Sequence |
| TRDistiller (with HHrepID) | | | |
| TRAL (STRs) | Multiple | Short tandem repeats | Sequence |
| TRAL (ARM) | | ARM repeats | |

The table lists the extent of the repeat prediction (single repeat or multiple repeat) and the criteria adopted by each method used in this work.

prediction methods noted above. The resulting subset of candidate proteins contains 16,431 entries (81% of the total).

The RBDmap database constructed by Castello and co-workers (ProteomeXchange accession number: PXD000883), corresponding to a comprehensive determination of RNA-interaction sites [28], contains 1,174 binding sites within 529 HeLa cell RNA-binding Proteins (RBPs) with a 1% false discovery rate (FDR). The experimentally determined RNA-binding peptides (RBDpeps) and computationally determined tandem repeat (tr) regions may not occur in the same part of the proteins. Thus, it is important to identify the overlap regions between the tandem repeats and RBDpeps, which we name trRBDpeps, in order to identify whether the repeated regions are involved in RNA-binding. For this work, we only applied the analysis to those proteins where at least 20% of their RBDpep residues are overlapped with tandem repeats.

We hypothesised that there might exist differences in the characteristics of the proteins identified depending on whether they have structural or sequence based repeats. The main reason for that intuition is founded on what is known about RNA binding and repeat sequences. For instance, previous works support the idea of disordered binding regions in proteins becoming structured after the attachment to RNA. These repeats are unlikely to be detected by structural methods and therefore we may be able to discern the relative abundance and importance of this type of RNA binding compared to well ordered RNA-binding repeats. With that perspective, we split the analysis into three different subsets: (i) The subset containing proteins only predicted as repetitive by sequence-based methods; (ii) Those containing only proteins with repetitive regions predicted based on structure-based methods; (iii) Those containing proteins detected as repetitive by both sequence and structure criteria.

To characterise the degree of disorder among the RBDpeps and the tandem repeat regions, for each protein in every subset we analysed the pLDDT score corresponding to the structure model predicted by AlphaFold and available in the AlphaFoldDB website [20, 21]. The pLDDT score represents a per-residue confidence metric included with the AlphaFold model. Regions with pLDDT > 90 are expected to be modelled with very high accuracy; regions with pLDDT between 70 and 90 are expected to be modelled well (a generally good backbone prediction); regions with pLDDT between 50 and 70 are predicted with low confidence; and regions with pLDDT < 50, suggest the region is either unstructured in physiological conditions or only structured as part of a complex [20, 21]. Recently, it was described that very low confidence pLDDT scores correlate with high propensities for intrinsic disorder [29].

To help interpret the results we constructed a visualisation for each protein showing the regions predicted as repetitive for every method, the experimentally determined RNA-binding peptides (RBDpeps), and the pLDDT score derived from the AlphaFold database (AlphaFoldDB) structure model predicted by AlphaFold (see S1 File). We use the same colours as the AlphaFoldDB website to represent structure prediction confidence corresponding to each pLDDT score. The whole analysis was performed differentiating the proteins predicted as repetitive by sequence, structure, or both of them.

The average percentage of the total protein length occupied by the tandem repeat regions and the RBDpeps were determined for every subset, as well as the pLDDT score profile corresponding to each of them. The pLDDT scores were then compared with those from the main isoforms of the whole human dataset of proteins.

According to previous works pointing to the role of certain residues as order or disorder promoters, and describing the enrichment of specific residues in RNA binding motifs, we hypothesised that the RBDpeps and tandem repeats possess different amino acid compositions compared to regions that are not repetitive or involved in RNA binding. According to that intuition, we calculated the amino acid composition within the tandem repeat regions as well

as for the RBDpeps, across all three subsets of proteins. We also calculated the amino acid composition of (i) regions that are not predicted as repetitive by any method and (ii) trRBDpeps, across all three subsets of proteins. We also calculated the amino acid composition for two different pLDDT score bins, (i) a pLDDT score <50, (ii) a pLDDT score ≥50, across all three subsets of proteins. Finally, we calculated the amino acid composition of all human proteins as reference.

We carried out a Gene Ontology (GO) molecular function term analysis of the subsets to find the most common functions associated with the tandem repeat RNA-binding protein. For the GO term analysis, we consider only the common terms present in at least 8 proteins among all 219 proteins of the three subsets.

To give further biological context to the trRBDpeps we have investigated their overlap with Pfam domains. We have taken annotations from Pfam 35.0 [30] and identified any regions that overlap by at least a single residue with our trRBDpeps. We have then aggregated these overlaps for each subset of proteins.

## Results

### Subsets overview

Our approach resulted in the identification of 219 RNA-binding tandem repeat proteins. They are all predicted as containing repeats by at least one method and are also present in the RBDmap database, with at least 20% of their RBDpeps overlapped with tandem repeats. We term these regions trRBDpeps (see Fig 3A). The list of these proteins is present in S1 Table of the S1 File. 136 proteins previously identified as RNA-binding [28] were not predicted to contain repeats by any method, or do not correspond to a main isoform (see S2 Table of the S1 File). Of the 219 RNA-binding tandem repeat proteins, 128 proteins were predicted as containing repeats by both sequence- and structure-based methods; 24 proteins were predicted to contain repeats only by sequence-based methods; and 67 proteins were predicted to contain repeats only by structure-based methods (Fig 3B). It is important to note that the number of trRBDpeps identified is limited by both the predictive power of the repeat detection methods employed, as well as the number of proteins included in the RBDmap. An example illustrating this limitation is the RNA-binding protein RO60 (UniProtKB: P10155), which is a previously characterised RNA-binding tandem repeat protein [31], but it is not present in the RBDmap. Consequently, it is not one of the proteins analysed in this work.

The coverage of RBDpeps and tandem repeats of every dataset as well as the overlap between them are summarised in Table 2. All subsets of sequences are covered by between one sixth and one quarter of their residues in RNA-binding peptides. The coverage of trRBDpeps in the sequence-based subset is lower than for the structure repeat proteins. This difference may be explained by the fact that the structure-based repeat set has a much higher coverage of repeats compared to the set of proteins uniquely found by the sequence-based methods. In both cases, the resulting average overlaps between those two regions are higher than the expected values taking into account the coverage of RBDpeps and tandem-repeats. The expected probability of overlap is 9.1% in both sequence-repeat proteins (Prob(TR) = 0.364 x Prob(RBDpep) = 0.250) and structure-repeat proteins (Prob(TR) = 0.576 x Prob(RBDpep) = 0.158). The observed overlaps are slightly higher than expected, with 10.8% (1.19 fold) and 11.3% (1.24 fold) for the sequence and structure subsets respectively (Table 2).

The proteins predicted as repetitive by both sequence and structure-based methods are those with the highest coverage of trRBDpeps. Besides, it is the subset with the highest coverage of tandem repeats on the average sequence length. The coverage of trRBDpeps (15.5%) is higher than the expected probability of overlap (1.24 fold), which is 12.5% (Prob(TR) = 0.616 x

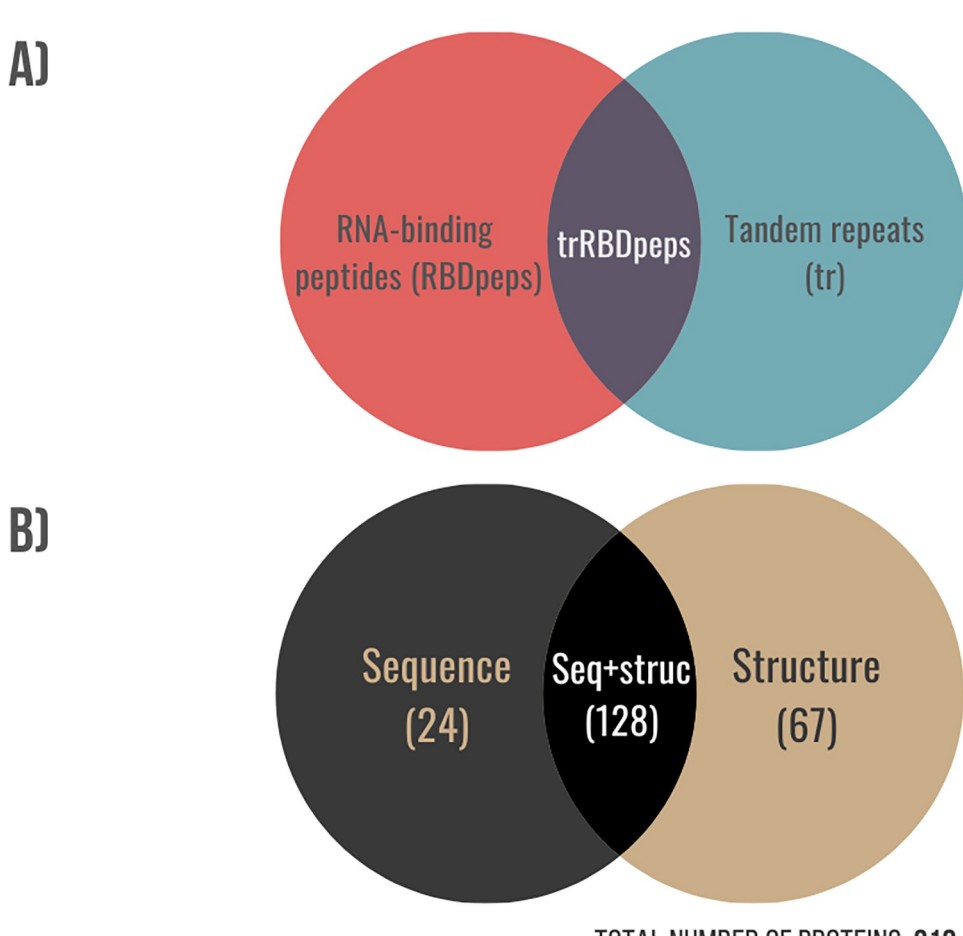

**Fig 3.** A) Venn diagram showing the different regions identified for proteins in this study. trRBDpeps are the intersection of residues between the tandem repeats and the RBDpep regions. B) Venn diagram showing the distribution of RNA-binding tandem repeat proteins identified by sequence and structural definitions of repeats.

Prob(RBD) = 0.203) for this subset. When we look at the proteins that have both sequence and structurally defined repeats we found overlap between sequence and structure repeats for 29.9% of the repeat region length. Thus, more than two thirds of the repeat regions are only defined by one type of method. The coverage of tandem repeats predicted by structural methods is higher than for sequence methods. One explanation is the ability of structure-based methods to identify repeats that have no sequence similarity. A representative case of this subset is shown in Fig 4.

**Table 2. Summary of the results concerning the coverage of RBDpeps and tandem repeats of every dataset, as well as the overlap between them.**

| Subset | Percentage coverage of tandem repeats | Percentage coverage of RBDpeps | Expected percentage coverage of trRBDpeps | Observed percentage coverage of trRBDpeps | Enrichment of observed vs expected coverage of trRBDpeps | Percentage coverage of tandem repeats predicted by both type of methods |
|---|---|---|---|---|---|---|
| Sequence repeats | 36.4 | 25.0 | 9.1 | 10.8 | 1.19 | - |
| Sequence + Structure repeats | 61.6 | 20.3 | 12.5 | 15.5 | 1.24 | 29.9 (of the 61.6%) |
| Structure repeats | 57.6 | 15.9 | 9.1 | 11.3 | 1.24 | - |

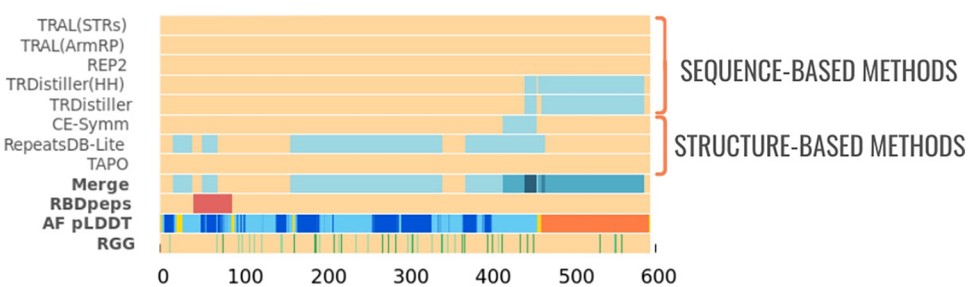

**Fig 4. An example of an RNA-binding tandem repeat protein (UniProtKB: O00567) predicted as repetitive by both types of methods.** It shows the larger coverage of repeats defined by structural methods, and the small overlap with the regions predicted as repetitive by sequence-based methods. The region corresponding to an RBDpep is highlighted in red. The "Merge" row corresponds to the union of the tandem repeats predicted by different methods. In the row corresponding to the pLDDT score, the segments are coloured according to the following scheme: orange (pLDDT< 50); yellow (50 >pLDDT< 70); light-blue (70 >pLDDT< 90); and blue (pLDDT> 90).

We observe that only a relatively low fraction (36%) of the sequence based subset is predicted as occurring in repeats. This could be because these proteins genuinely possess less repeats, or it could be that there are more repeats but these are too divergent for the sequence methods to detect. For this set, the repeats found are detectable by sequence but not by structure. This would be expected if these repeats either had no regular structure or were organised in non-symmetrical ways that were not detected by structure based methods. Previous evidence suggests that the binding process between disordered regions and RNA molecules orient their transition to well-ordered conformations [19, 32, 33]. These cases may not be detected by structure-based methods designed for the detection of structurally regular repeats.

## pLDDT profiles

In this section we explore the distribution of pLDDT scores for each subset generated in previous sections (Fig 5). The panels to the left correspond to the distributions of each subset considering the whole proteins. The panels to the right only consider the regions corresponding to trRBDpeps.

Among the sequence-based subset, both tandem repeats and the regions that are not predicted as repetitive are enriched in likely disordered residues. The trRBDpeps distribution of this subset shows a bimodal distribution with peaks at pLDDT < 50 and at 70 > pLDDT < 90 (Fig 5B). Since this subset is the smallest among all the ones studied in this work, the results may be heavily influenced by just a single protein. The LRPPRC protein (UniProtKB: P42704) represents a high proportion of the total trRBDpeps in this subset, with almost all of them located in regions with a pLDDT score between 70 and 90 (see Examples of novel trRBP and S1 File). This long protein introduces a bias on the pLDDT score distribution of the whole subset.

On the contrary, proteins present in the structure-based subset are highly enriched in likely ordered residues (pLDDT >50) (Fig 5E). 97.0% of the residues in tandem repeat regions in this subset have a pLDDT >50, and more than 93.5% of the RBDpeps are located in regions with a pLDDT score >50. This tendency is not observed in the sequence-based dataset, where

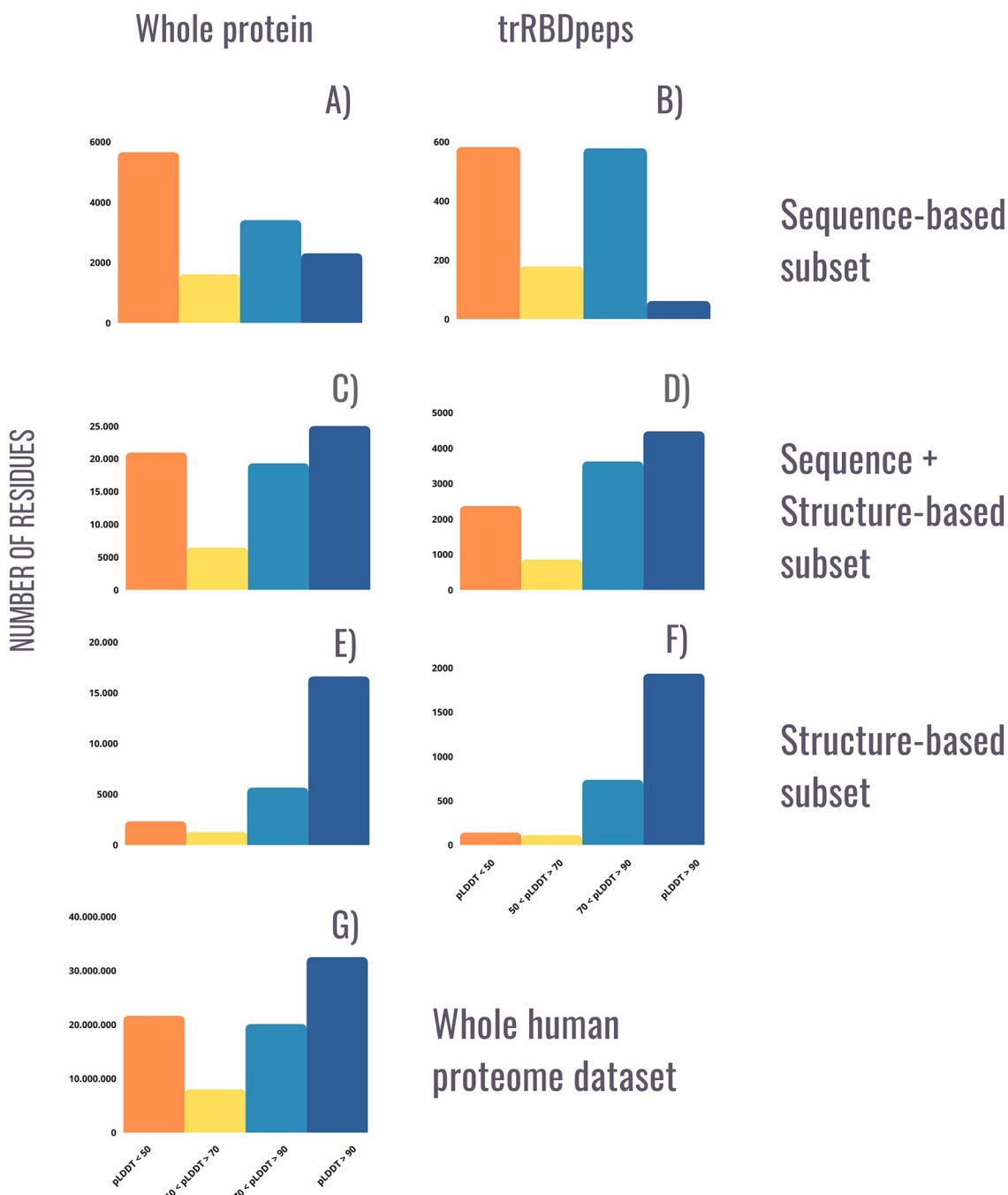

**Fig 5. Distribution of pLDDT scores across the 219 RNA-binding tandem repeat proteins, split by repeat prediction method and region (whole protein or trRBDpeps).** As in the AlphaFoldDB website, the orange bar corresponds to the amount of residues among the dataset with a pLDDT score <50; the yellow bar corresponds to a pLDDT score between 50 and 70; the light-blue bar corresponds to a pLDDT score between 70 and 90; and the blue bar corresponds to values >90. A) pLDDT score profile for the whole set of RNA-binding tandem repeat proteins predicted as repeated by sequence-based methods. B) pLDDT score profile for trRBDpeps of proteins predicted as repeated by sequence-based methods. C) pLDDT score profile for the whole set of RNA-binding tandem repeat proteins predicted as repeated by both types of methods. D) pLDDT score profile for trRBDpeps of proteins predicted as repeated by both sequence and structure-based methods. E) pLDDT score profile for the whole RNA-binding tandem repeat proteins subset predicted as repeated by structure-based methods. F) pLDDT score profile for trRBDpeps of proteins predicted as repeated by structure-based methods. G) Distribution of pLDDT scores across the whole human dataset (20,264 proteins).

only 39.3% of the RBDpeps are placed in ordered regions. The subset corresponding to the proteins predicted as repetitive by both types of methods shows an intermediate situation, where 69.2% of the RBDpeps are located on regions with pLDDT >50. A similar trend can be seen by comparing the trRBDpeps regions of these subsets (Fig 4B–4F). While in the structure-based subset 95.2% on average of trRBDpeps are placed in regions with a pLDDT score >50, the same value is 58.4% in the sequence-based dataset.

The subset formed by proteins predicted as repetitive by both types of methods represents an intermediate situation, where 79.1% on average of the trRBDpeps are located in regions with a pLDDT score >50 (Fig 5D). This set also has the highest fraction of residues (15.5%) predicted to be in trRBDpeps, and more than 86.4% of the tandem repeat regions corresponds to a pLDDT score >50 in this subset.

It is useful to compare the pLDDT profiles of the RNA-binding tandem repeat protein with the total set of 20,264 human proteins (Fig 5G). The proteins predicted as repetitive by sequence methods are more enriched in regions with low pLDDT scores (pLDDT<50), contrary to the structural subset with few residues with low pLDDT scores. The proteins predicted as repetitive by both types of methods present the most similar pattern to that of the human proteome set (Fig 5C).

## Amino acid composition

RNA-binding proteins and RNA-binding regions have biases in the amino acids that encode their primary sequences. We decided to explore the amino acid compositions of the RNA-binding proteins split across our three subsets to see if we could find differences between them. First, we investigated whether there was a compositional difference between the high and low pLDDT regions of the 219 RNA-binding tandem repeat proteins (Fig 6A). To understand whether these differences are specific to RNA-binding repeat proteins, we calculated the amino acid enrichment of the trRBDpeps with respect to the whole human dataset (Fig 6C), and compared this to the distribution across all human proteins (Fig 6B).

Our results show that glycine, followed by serine and proline are the most strongly enriched amino acids in the low pLDDT regions of the RNA-binding repeat proteins (Fig 6A). A similar tendency is observed among the whole human proteome as shown in Fig 6B, with the addition of alanine and threonine also being enriched in low pLDDT regions. These results are similar to previous studies, where proline, serine, arginine, and glycine were postulated as disorder-promoting residues enriched in RNA-binding proteins [28].

An example of how the glycine residues tend to be positioned on low pLDDT regions is presented in Fig 7, where these patches form islands often flanking globular domains in the TAF15 protein (UniProtKB: Q92804). This pattern was previously described as a functional cooperation between natively structured and unstructured regions [17]. In the context of this protein, a consistent GGYDR consensus sequence repeat is observed among the residues proximal to the C-terminal region. The consistency and frequency of this particular motif show that this represents a potentially novel bona fide repeat rather than the result of a biased sequence composition.

The RNA-binding tandem repeat regions with pLDDT >50 present a high proportion of leucine, isoleucine, lysine, phenylalanine and valine, which is expected since hydrophobic residues are commonly conserved within globular domains [34]. A higher proportion of cysteine residues is observed within ordered regions. This result is consistent with previous works, where cysteine has been noted as a "order-promoting" residue [35]. Similar tendencies are shown in Fig 6B concerning the whole human proteome dataset. However, the details do differ, with tyrosine being enriched in regions with pLDDT >50, which is not observed among trRBDpeps subsets, and lysine being less enriched than in those cases.

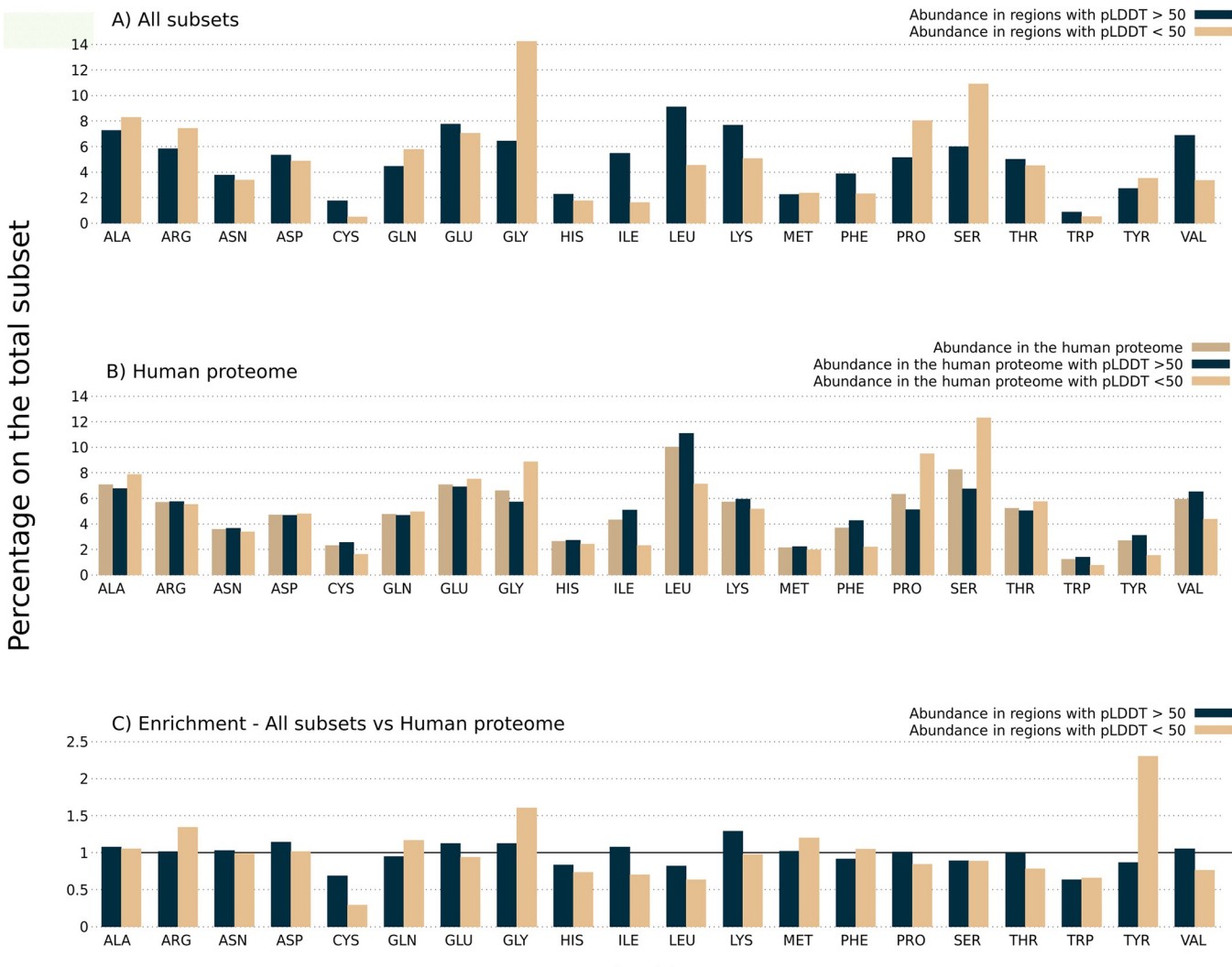

**Fig 6.** A) Amino Acid composition in regions with low and high pLDDT scores among the 219 RNA-binding tandem repeat proteins. B) Amino acid composition in regions with low and high pLDDT scores among the whole human proteome dataset, as well as for the complete sequences. C) Amino acid enrichment in regions with low and high pLDDT scores among RNA-binding tandem repeat proteins with respect to the whole human dataset. All values above 1 indicate an amino acid enrichment in RNA-binding tandem repeat proteins compared to the human proteome dataset. The opposite holds true for values below 1.

To better understand the differences between the tandem-RNA-binding repeat proteins and the whole human proteome we look at the enrichment of amino acid composition at both high and low pLDDT scores (Fig 6C). In the low pLDDT regions we see an enrichment of arginine (1.34 fold), glycine (1.61 fold) and tyrosine (2.30 fold) among the 219 RNA-binding tandem repeat proteins. The highest fold enrichment of tyrosine in low pLDDT regions is consistent with previous works that describe how RBDpeps represent hotspots for tyrosine phosphorylation [28]. The RGG boxes formed by enriched glycine and arginine residues were previously reported to form peptides positioned along the major groove of RNA-protein complexes, by recognizing duplex-quadruplex junctions [19]. This motif is also commonly associated with disordered regions [18]. By its side, flexible regions in RNA-binding proteins rich in serine and arginine, as well as arginine and glycine were previously postulated as contributors to the interaction with nucleic acids [36, 37]. Particularly, disordered repeat motifs formed by

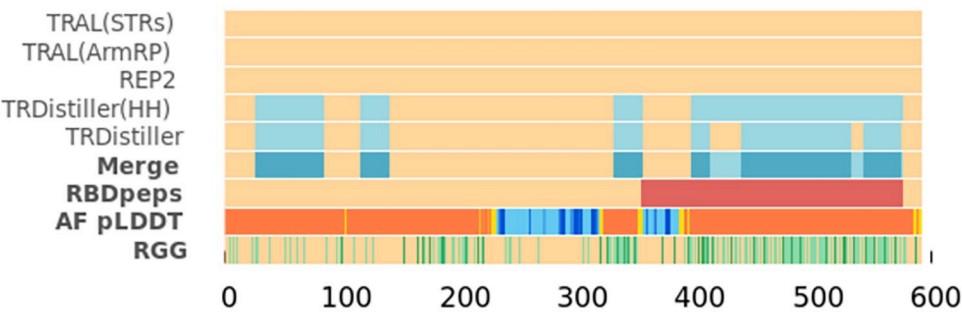

**Fig 7. An example of a protein (TAF15, UniProtKB: Q92804) showing how the glycine residues of the RGG motif tend to be positioned in low pLDDT regions (RGG row, highlighted in green).**

arginine and serine were reported as contributors in export, translation, stabilisation and splicing of RNA molecules [18,38, 39].

In the high pLDDT regions of RNA-binding tandem repeat proteins, lysine is the only strongly enriched amino acid (1.29 fold). Lysine is well known as a residue enriched in RNA-binding sites from numerous crystal structures. It is interesting to see the different preference of lysine and arginine to appear in high versus low pLDDT regions in RNA-binding proteins despite their chemical similarity.

To identify whether there are differences between the different subsets regarding their amino acid composition in RBDpeps and non-RNA-binding regions, we now focus on the amino acid composition of the RBDpep regions among all subsets (Fig 8). The differences between the subsets concerning this particular aspect can be then contextualised with the different biological roles they may play, as discussed in the next section of this work.

All the subsets show a high proportion of glycine in RBDpeps with respect to non-RNA-binding regions. However, that tendency is not as strong among the proteins predicted as repetitive by structure-based methods (Fig 8C), presumably because the enrichment of glycine is largely due to the presence of RGG repeats in the disordered regions. In addition, we see that the structure-based subset is the only one with a higher proportion of glutamic acid and histidine and presents a low proportion of arginine, asparagine, aspartic acid, phenylalanine, tryptophan, and tyrosine among the RNA-binding regions. This subset also shows a higher composition of hydrophobic amino acids in RBDpeps, such as alanine and valine, but also a higher proportion of lysine.

The proteins predicted as repetitive by sequence-based methods show the highest proportion of arginine, cysteine, aspartic acid and serine among RBDpeps (Fig 8A). On the contrary, alanine, lysine, glutamine, glutamic acid, and proline are somewhat depleted. The proportion of tyrosine in RNA-binding regions is similar to the subset composed by proteins predicted as repetitive by both types of methods (Fig 8B). This subset represents an intermediate situation,

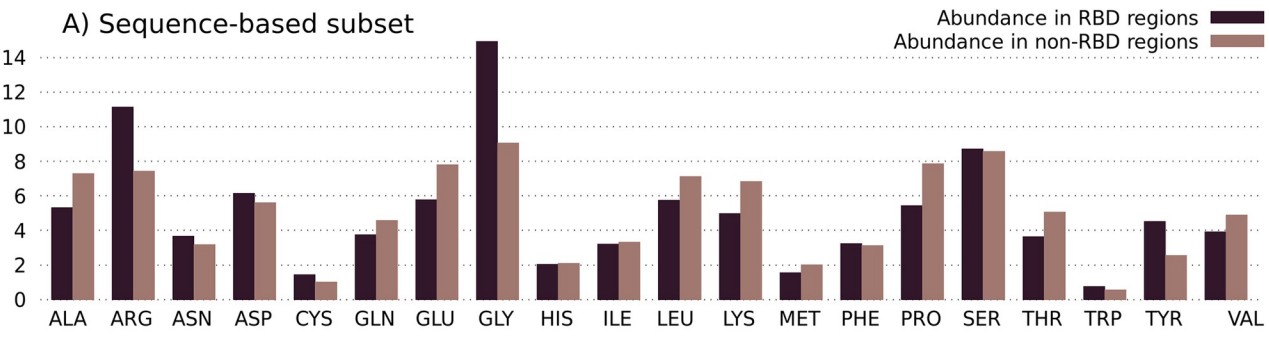

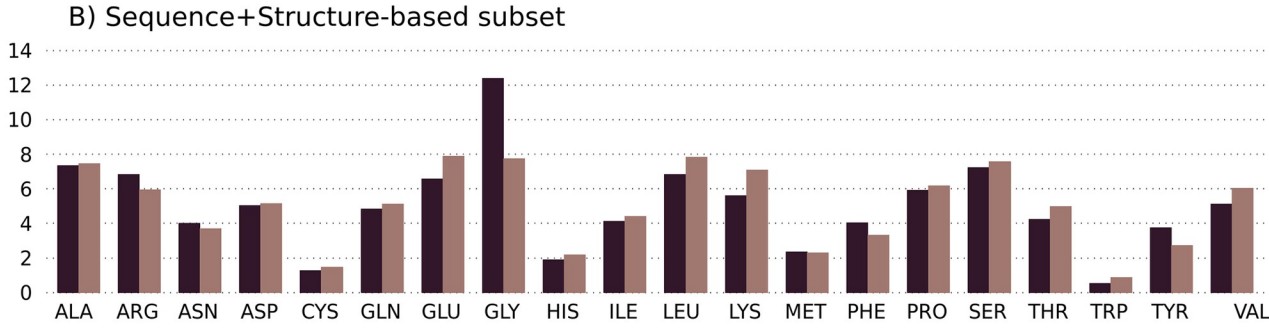

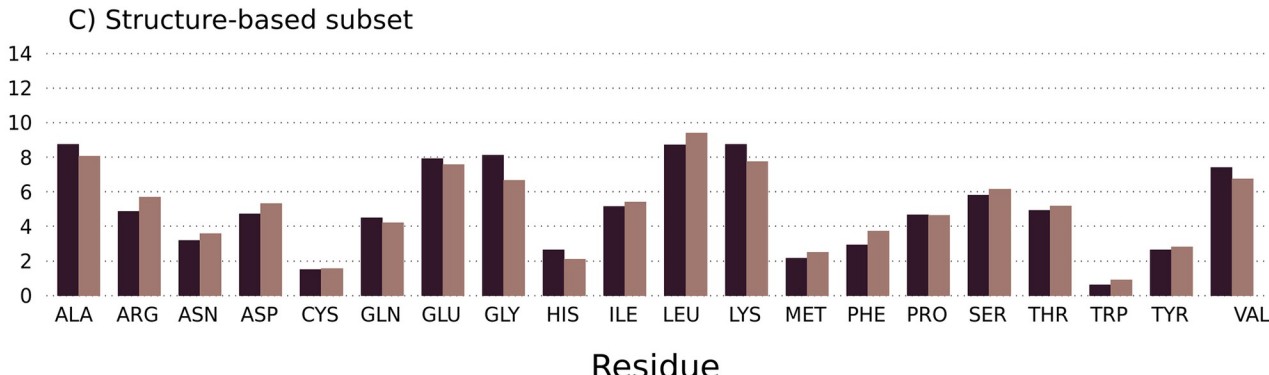

**Fig 8. Amino acid composition in RBDpeps and non-RBDpeps among RNA-binding tandem repeat proteins.** A) Amino Acid composition among the RNA-binding tandem repeat proteins predicted as repetitive by sequential methods. B) Amino Acid composition among the RNA-binding tandem repeat proteins predicted as repetitive by both types of methods. C) Amino Acid composition among the RNA-binding tandem repeat proteins predicted as repetitive by structural methods.

since it is subtly enriched in arginine and asparagine within RBDpeps with a low proportion of glutamine, glutamic acid, lysine, proline and valine, such as in the sequence-based subset, but the proportions of aspartic acid, serine and tryptophan show a tendency similar to the structure-based subset.

To complete the analysis, we explored whether there exist differences concerning the amino acid composition of tandem repeat regions. An enrichment in glutamic acid and lysine can be observed with respect to non-repeated regions among the proteins predicted as repetitive by sequence methods (Fig 9A). It was previously reported that multiple lysine may form

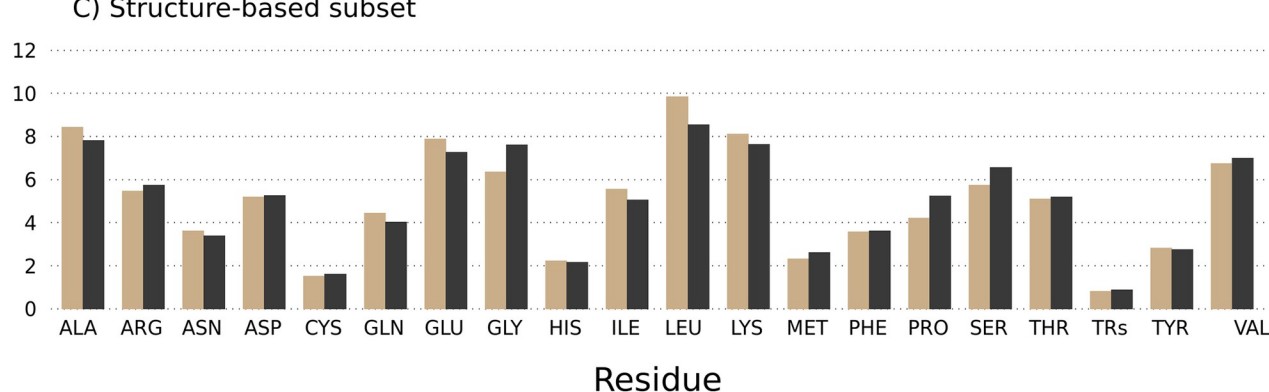

**Fig 9. Amino acid composition in tandem repeats and non-repeated regions.** A) Amino Acid composition among the RNA-binding tandem repeat proteins predicted as repetitive by sequential methods. B) Amino Acid composition among the RNA-binding tandem repeat proteins predicted as repetitive by both types of methods. C) Amino Acid composition among the RNA-binding tandem repeat proteins predicted as repetitive by structural methods.

disordered regions that interact with RNA [18]. The abundance of arginine is also present particularly in this specific subset. The same holds true for leucine in the structure-based subset, and also among those proteins predicted as repetitive by both types of methods (Fig 9B and 9C, respectively). These results are similar to those corresponding to the original set of 20,264 proteins (S1 Fig in S1 File). A small enrichment in histidine can also be observed among repetitive and structured regions in all subsets.

## GO term analysis

For each subset we performed a GO term analysis in order to give a biological context to the RNA-binding tandem repeat proteins identified. Fig 10 summarises the results of our GO term analysis. The radius of the circles are calculated to be proportional to the number of GO terms matched. We then subsequently normalise the radii of all circles by the number of proteins of each subset to make each set comparable.

Unsurprisingly, the GO term corresponding to "RNA-binding" (GO:0003723) is consistently the most frequent in all the subsets. However, by looking in more detail at the results, we observe differences with respect to the specific molecules bound by the proteins present in the different subsets. For example, the terms "mRNA-binding" (GO:0003729) and "mRNA-3'-UTR-binding" (GO:0003730) are underrepresented in the structure repeat set compared to the others. This is suggestive that binding sites for mRNAs are enriched in disordered regions of these proteins. As an example the FUS protein (UniProt:P35637) which contains an RRM and a RanBP type zinc finger has an RBDpep that overlaps with a C-terminal disordered region enriched in RGG repeat motifs. We find the same pattern of C-terminal RGG repeat motifs in Transformer-2 protein homolog beta (UniProt:P62995). Indeed, many of the proteins in this set contain RGG motifs that likely explain the RNA binding property of the RBDpep.

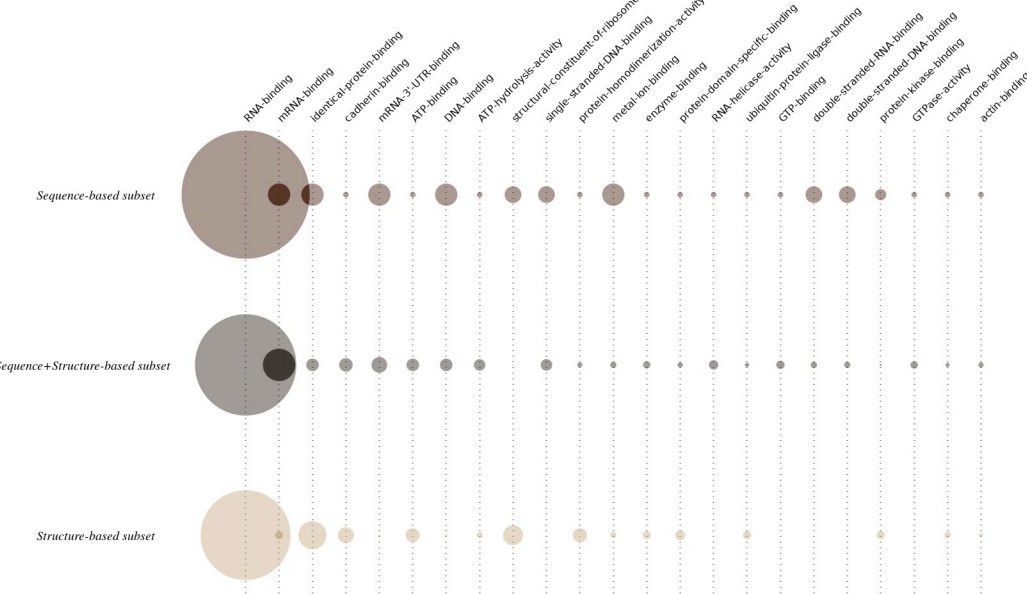

**Fig 10. Circle representation of the GO terms identified in every subset studied in this work.** Every column corresponds to one of the GO terms present in at least 8 entries in the whole RNA-binding repeat protein dataset. Each row corresponds to one of the subsets studied in this work. The radius of a circle is proportional to the number of matches for a specific GO term in each subset. Then all circle radii are further normalised in each set by the total number of proteins in each set to make them comparable.

The terms "DNA-binding" (GO:0003677), "single-stranded DNA-binding" (GO:0003697), "double-stranded DNA-binding" (GO:0003690), and "double-stranded RNA-binding" (GO:0003725) GO terms are enriched among proteins predicted as repeated by sequence-based methods. The term "metal ion binding" (GO:0046872) is also over-represented among the sequence repeat proteins with respect to the remaining subsets. This may be due to the relative abundance of zinc-binding domains and motifs in each subset which will be discussed in the next section.

The GO terms "identical protein binding" (GO:0042802) and "protein homodimerization activity" (GO:0042803), are enriched among the proteins predicted as repeated by structure based methods, suggesting that oligomeric binding of RNA-binding repeat proteins is mediated by the ordered parts of the proteins. We also see a similar pattern of enrichment in the structured repeat proteins for the GO terms "ATP-binding" (GO:0005524), "cadherin binding" (GO:0045296), "enzyme binding" (GO:0019899), "protein domain specific binding" (GO:0019904), "structural constituent of ribosome" (GO:0003735) and "ubiquitin protein ligase binding" (GO:0031625).

As a whole, this analysis shows that although the common characteristic throughout all the subsets is the binding to RNA molecules, there are biological characteristics that are specific to each subset.

## Overlap with Pfam domains

To complement the analysis presented so far, we now study which Pfam domains are located in the same regions where trRBDpeps are found. These results should in principle help to identify tandem repeat RNA-binding domains like the Pumilio Family. An abbreviated version of the results of our analysis is shown in Table 3, where the column titled "trRBDpep matches" refers to the number of tandem repeat RBDpeps that match with Pfam domains among a certain subset. The full list of all the trRBDpep matches with Pfam domains is presented in S3 Table in S1 File.

Among the proteins predicted as repetitive by structure or by both sequence and structure, the main Pfam domain identified is the well known RRM domain (Pfam: PF00076). However, among the sequence-repeat proteins, the domain found most commonly overlapped with trRBDpeps is the Zinc knuckle (Pfam: PF00098), which is often found to bind to nucleic acids [40]. Unexpectedly, the Tau and MAP protein tubulin-binding repeat (Pfam: PF00418) is the second most common domain with three repeat occurrences overlapping RBDpeps found in Tau (UniProt:P10636) and Microtubule-associated protein 4 (UniProt:P27816). According to the Pfam alignment this repeat contains two conserved lysines and a GGG motif which is reminiscent of the RGG motifs. Tau protein is important in several human diseases including Alzheimer's disease. Aggregates of the Tau protein have recently been shown to include noncoding RNAs [41] enriched in snRNAs and snoRNAs. We suggest that these C-terminal repeat regions of Tau may mediate some of these interactions.

Among the proteins predicted as repeated by both sequence and structure, the second most common Pfam domain matching trRBDpeps is the previously reported RNA-binding tandem repeat KH domain (Pfam:PF00013) [2, 3], while the third one is the DEAD/DEAH box RNA helicase (Pfam:PF00270). The DEAD/DEAH domain is involved in clamping and unwinding RNA molecules in a variety of biological processes [42]. The same domain is also present in adaptor proteins acting as cytosolic DNA sensors, or as regulators of signalling and gene expression [43]. DEAD/DEAH box helicases are composed of two core domains, with the Helicase C domain (Pfam:PF00271) found to the C-terminus of the DEAD/DEAH box helicase domains. These two domains are highly similar in terms of their structure because

**Table 3. Abbreviated list of the Pfam domains located in trRBDpeps among the subsets studied in this work.**
Only entries with more than one trRBDpep match per Pfam family are shown. The complete results can be found in S3 Table in S1 File.

| trRBDpep matches | Pfam code | Description |
|---|---|---|
| **Sequence-based subset (24)** | | |
| 6 | PF00098 | Zinc knuckle |
| 3 | PF00418 | Tau and MAP protein, tubulin-binding repeat |
| 2 | PF01535 | PPR repeat |
| **Sequence+Structure-based subset (128)** | | |
| *trRBDpap matches* | | |
| 76 | PF00076 | RNA recognition motif. (a.k.a. RRM RBD or RNP domain) |
| 27 | PF00013 | KH domain |
| 5 | PF00270 | DEAD/DEAH box helicase |
| 3 | PF00565 | Staphylococcal nuclease homologue |
| 3 | PF00271 | Helicase conserved C-terminal domain |
| 2 | PF13516 | Leucine Rich repeat |
| 2 | PF09011 | HMG-box domain |
| 2 | PF08080 | RNPHF zinc finger |
| 2 | PF02218 | Repeat in HS1/Cortactin |
| 2 | PF01424 | R3H domain |
| 2 | PF01271 | Granin (chromogranin or secretogranin) |
| 2 | PF00880 | Nebulin repeat |
| 2 | PF00505 | HMG (high mobility group) box |
| 2 | PF00458 | WHEP-TRS domain |
| 2 | PF00400 | WD domain G-beta repeat |
| 2 | PF00307 | Calponin homology (CH) domain |
| 2 | PF00041 | Fibronectin type III domain |
| **Structure-based subset (67)** | | |
| *trRBDpep matches* | | |
| 7 | PF00076 | RNA recognition motif. (a.k.a. RRM, RBD, or RNP domain) |
| 4 | PF01248 | Ribosomal protein L7Ae/L30e/S12e/Gadd45 family |
| 3 | PF00244 | 14-3-3 protein |
| 3 | PF00085 | Thioredoxin |
| 2 | PF00056 | lactate/malate dehydrogenase, NAD binding domain |
| 2 | PF00012 | Hsp70 protein |

they have both derived from an ancient duplication event. The *Staphylococcal* nuclease homologous domain (Pfam:PF00565) is also identified in the list. We also find Pfam domains annotated with the Repeat type (All Pfam entries are assigned to one of six types: Family, Domain, Repeat, Motif, Coiled-coil or disordered). These include the Leucine Rich repeat (Pfam: PF13855) [44], the WD40 domain G-beta repeat (Pfam:PF00400) [45], and the Nebulin repeat (Pfam:PF00880) [46].

The second ranked Pfam domain matching with tandem repeat RBDpeps among the structure-based subset is the Ribosomal protein L7Ae/L30e/S12e/Gadd45 family (Pfam: PF01248). This family includes ribosomal proteins (S12, L30e), proteins binding guiding RNA (L7Ae, 15.5 kD, fibrillarin), as well as components of ribonuclease P [47]. These proteins bind functionally diverse RNAs, including ribosomal RNA, snoRNA, snRNA and mRNA.

Within the structure subset we see many metabolic enzymes which at first seems surprising. For example, Fructose-bisphosphate aldolase, Isocitrate/isopropylmalate dehydrogenase and

D-isomer specific 2-hydroxyacid dehydrogenase. However, the work of Hentze and others [48, 49] has shown that many core metabolic enzymes can bind to RNA. These enzymes may bind to their own mRNA to regulate the level of translation.

Beyond the most frequent entries of each subset, we can observe how the Pfam domains completing the tables are highly related with their common characteristics. Thus, domains associated with the interaction with different RNA molecules as the ribosomal protein L19e (Pfam: PF01280) [50] and the 60s Acidic ribosomal protein (Pfam: PF00428) [51] are represented in the sequence-based subsets, as for domains identified as repeats, such as the PPR repeat (Pfam: PF01535) [52]. The structure-based subset show the Ribosomal protein 50S L24/ mitochondrial 39S L24 (Pfam: PF17136) [53] and the tRNase Z endonuclease domain (Pfam: PF13691) [54], as several domains related to the interaction with other proteins and enzymes like with 14-3-3 protein (Pfam: PF00085) [55] and the Ubiquitin carboxyl-terminal hydrolases (Pfam: PF18031) [56], in agreement with the results presented in the previous section. This observation suggests that the tandem repeat RBDpeps may also play a dual role in protein-protein and protein-enzyme interactions among structure-repeated proteins. Finally, the subset corresponding to the proteins predicted as repetitive by both sequence and structure shows characteristics related to the previous subsets. This subset contains WD40 repeats and the KH domain, as well as protein:protein and protein:enzyme interaction domains such as the calponin homology (CH) domain (Pfam: PF00400) [56, 57]. This is the subset with the higher number of matches with Pfam repeat domains, consistent with containing the highest fraction of residues predicted to be in repeats.

## Examples of novel trRBP

**A novel trRBP based on sequence.**   The Serine/Arginine-rich splicing factor 2 (SRSF2) protein, is not annotated as repetitive in UniProt. In previous works [58, 59] it was reported as a protein necessary for the splicing of pre-mRNA as well as for the formation of the earliest ATP-dependent splicing complex. It is also capable of forming interactions with spliceosomal components, in agreement with the RNA-binding analysed here. In this work, this protein is only predicted as repetitive by sequence-based methods. The prediction shows multiple repetitive regions, with two of them overlapping with the RBD of the protein. A consistent RS repeat is observed by examining the sequence of the RBD overlapped with the region predicted as repetitive by the largest number of methods (See Fig 11). This region is predicted as disordered by AlphaFold, thus showing an example of a trRBP where a bona fide repeat is only detected by its sequence.

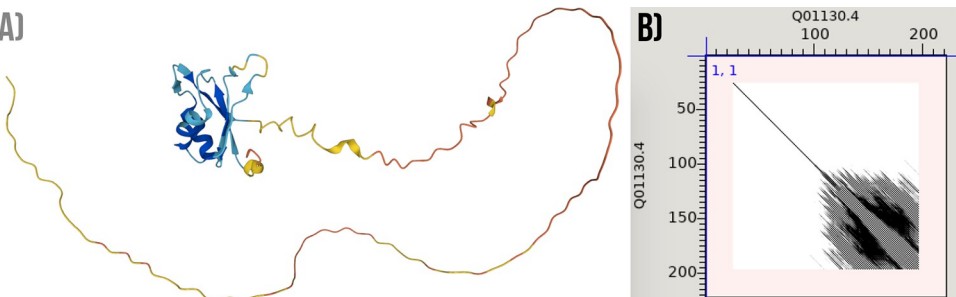

**Fig 11. An example of a protein (SRSF2, UniProtKB: Q01130) previously described as RNA-binding but not yet annotated as repetitive.** A) Structural visualisation coloured according to the pLDDT score scheme: orange (pLDDT< 50); yellow (50 >pLDDT< 70); light-blue (70 >pLDDT< 90); and blue (pLDDT> 90). B) Dot plot representation of the protein sequence aligned against itself, showing that a consistent pattern of repetition overlaps with its disordered region (residues 117–191).

**A novel trRBP based on sequence and structure.** The FAM98A protein (Family with sequence similarity 98, member A), positively stimulates PRMT1-induced protein arginine methylation and is involved in promoting colorectal cancer cell malignancy [60]. It was previously annotated in UniProt by its RNA binding function but has not been listed as a repetitive protein. However, we found it is predicted as having repeats by both sequence and structure-based methods. Both RepeatsDB Lite and CE-Symm detect structural repeats in this protein, but in different regions (See Fig 12): while RepeatsDB Lite predicts, according to its own classification, a fibrous repeat consisting of an extended alpha helix among the residues 198 to 233, CE-Symm detects a symmetrical arrangement of alpha helices within the range 1–137. On the other hand and on the basis of the protein's sequence, TRDistiller predicts repeat regions in three stretches close to the C-terminal region (residues 334–388, 407–426, and 460–474). In agreement with previous observations, this region of the protein is also predicted as being highly disordered. By examining the sequence it is possible to detect two consistent RGG boxes, one of them located around residues 350 to 380 and the other between residues 457 to 484. Those two regions are the ones directly involved in RNA binding. It is also interesting to note a consistent GGY motif repeated several times among the residues 391 to 441. These sequentially-determined tandem repeats are also evident in a dot plot representation (S2 Fig in S1 File).

**A novel trRBP based on structure.** The GTP-binding protein 4 (GTPBP4) is Involved in the biogenesis of the 60S ribosomal subunit [61]. Despite it not previously annotated as a repetitive protein, both RepeatsDB Lite and CE-Symm detect tandem repeats in this example. Both methods agree in predicting a tandem helical repeat close to the N terminus (Fig 13). This is the motif that interacts with the target RNA molecule. However, each of the methods also predict one additional tandem repeat: while RepeatsDB Lite detects an Alpha/Beta solenoid between residues 350 and 400, CE-Symm predicts a symmetrical arrangement between residues 250 and 320.

**An example of a false negative prediction: A structural trRBP only detected by sequence-based methods.** The Leucine-rich PPR motif-containing protein (LRPPRC) has been previously reported as repetitive, and its role in RNA metabolism for both nuclei and mitochondria was also discussed [62]. In the particular context of this work, only the sequence-based TRDistiller method in both versions can predict parts of this protein as

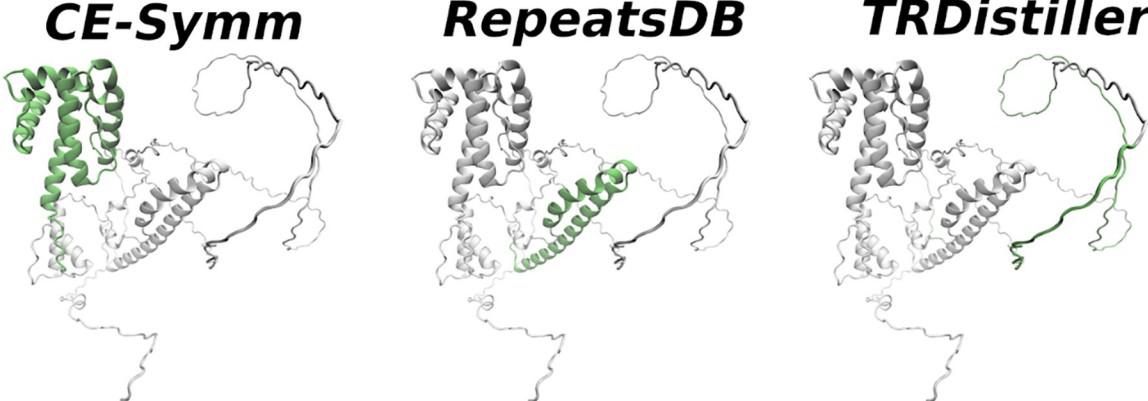

**Fig 12. An example of a protein, FAM98A (UniProtKB: Q8NCA5), previously described as RNA-binding but not yet annotated as repetitive.** The green representation highlights the regions predicted as tandem repeats by the methods mentioned above each structure. CE-Symm detects a symmetrical arrangement of alpha helices within the residues 1 to 137; RepeatsDB Lite detects, according to its own classification, a fibrous repeat helix among the residues 198 to 233; TRDistiller predicts a repeat in three discontinuous disordered stretches from residues 334 to 474, close to the C-terminal region.

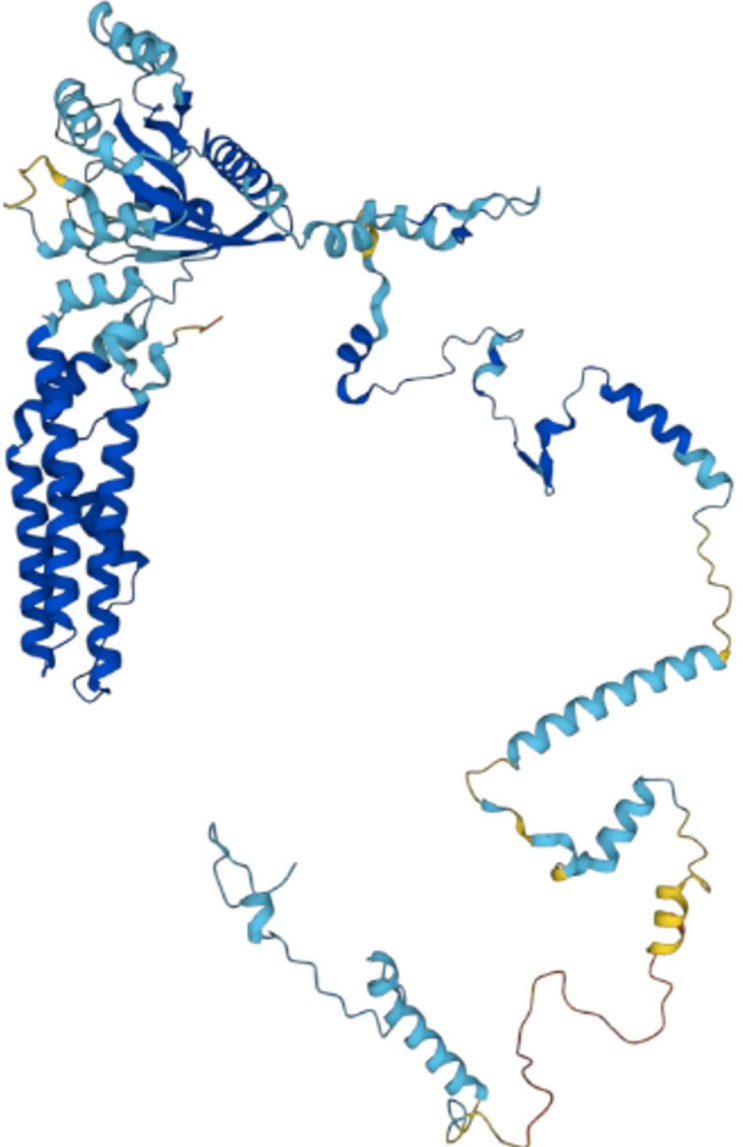

**Fig 13. An example of a protein (GTPBP4, UniProtKB: Q9BZE4) previously described as RNA-binding but not yet annotated as repetitive.** The structural visualisation is coloured according to the pLDDT score scheme: orange (pLDDT< 50); yellow (50 >pLDDT< 70); light-blue (70 >pLDDT< 90); and blue (pLDDT> 90). Both CE-Symm and RepeatsDB-Lite detect a tandem repeat close to the N terminal region (left side of the representation), forming a TIM-barrel motif according to the RepeatsDB classification.

repetitive. By observing the AlphaFold model for this protein (Fig 14A), it is possible to observe a clear structural pattern forming a symmetrical extended repeat. A consistent patch of positively charged residues is present among the structural tandem repeats (Fig 14B), coinciding with the region binding RNA. It is interesting to note that this structural tandem repeat can be detected by the algorithm REP2 by relaxing the default cut-off criteria, thus suggesting that the parameters employed with the methods used in this work may be too strict in some contexts. This example is a good case of a false negative concerning the detection of tandem repeats, particularly those with a structural pattern besides the sequence similarity.

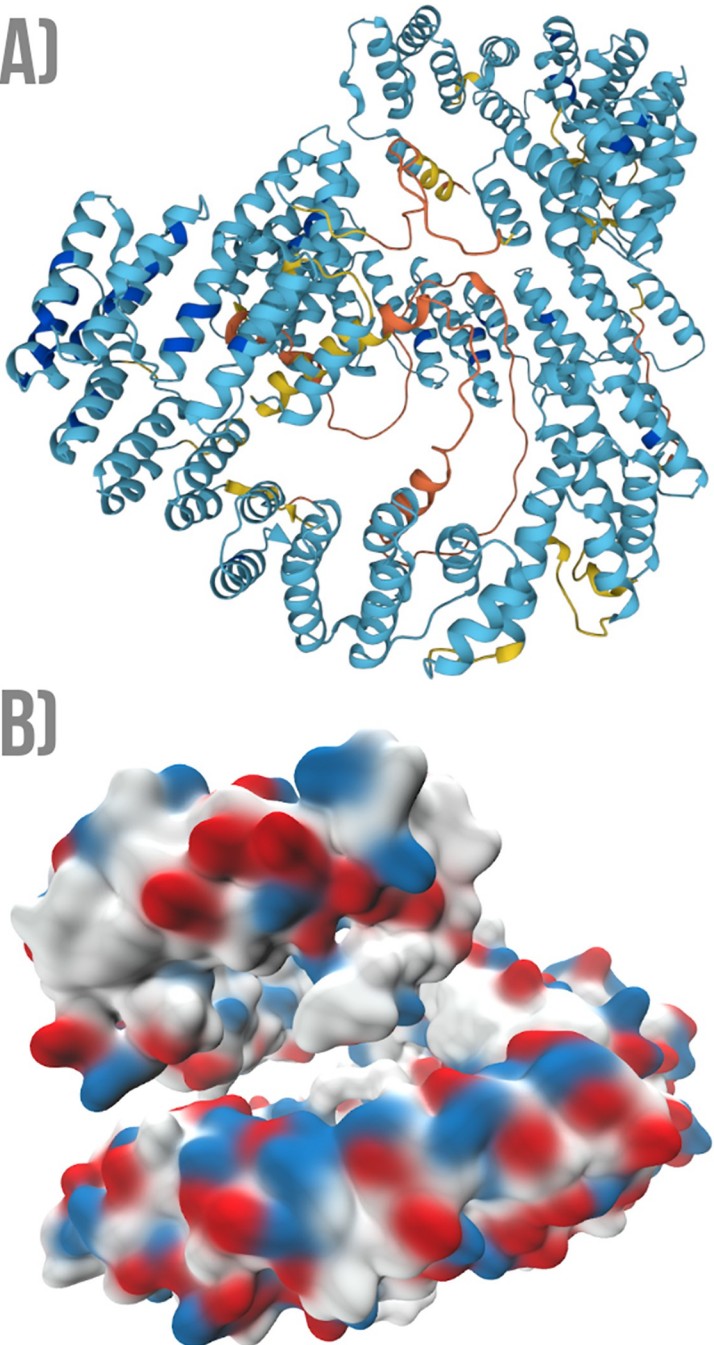

**Fig 14. An example of a protein (LRPPRC, UniProtKB: Q9BZE4) forming a repetitive structural pattern that is only detected by sequence-based methods.** A) Structural visualisation coloured according to the pLDDT score scheme: orange (pLDDT< 50); yellow (50 >pLDDT< 70); light-blue (70 >pLDDT< 90); and blue (pLDDT> 90). B) Van der Waals surface representation, where the negatively charged residues are highlighted in red, while those with a positive charge are highlighted in blue.

## Discussion

We identified and characterised 219 RNA-binding tandem repeat proteins. Among them, we differentiate between the proteins predicted as repetitive by sequence-based methods (24 entries) and those containing structural-tandem repeats (67 entries). We also analysed the

proteins predicted as repeated by both types of methods (128 entries). We observed that 81% of the main isoforms from the whole human proteome (20,206 proteins) were predicted as repetitive by at least one of those methods. Thus, having the AlphaFold models for the whole human proteome has greatly increased our ability to identify proteins with tandem repeats within them.

Can all tandem repeats be identified using structure alone? The answer is clearly not, because the sequence-based methods can predict repeats even in disordered regions that are not accessible for structure-based methods. Consequently, we see an enrichment of likely disordered residues among the proteins predicted as repetitive by sequence-based methods only. The proteins in the sequence only subset are generally enriched in likely disorder even outside the tandem repeat or RBDpep regions.

We found that the proteins identified as repetitive based on structural predictions only, show a higher coverage of tandem repeats. This may be because the structural methods can identify repeats that have little or no sequence similarity, while the sequence-based methods rely on significant sequence similarity being detected to define tandem repeats. As may be expected, the average percentage of the total RBDpeps length overlapped with tandem repeats is higher in structure-based repeats since the portion of the proteins occupied by RBDpeps is similar in all the subsets studied.

Our results suggest that highly-structured repeats are perhaps the most significant contributors to the RNA binding process, but the disordered repeated regions may play an important complementary role. For example, they can act as targets for post-transcriptional modifications preferentially found in intrinsically disordered regions [63], thus influencing the binding process itself or causing local structural changes [18]. It is interesting to remark that the sequence-based subset is the one with the highest abundance of arginine and glycine residues, with the RGG box being one of the typical targets for post-translational modifications [18]. It has been previously postulated that a combination of classical RNA-binding domains and disordered regions may function cooperatively [17, 18]. Recent evidence suggests that disordered regions emerge as frequent RNA interaction sites in vivo [25]. We often saw disordered trRBDpeps are adjacent to classical RNA-binding domains.

We found that sequence and structure-based predictors mostly identify different regions of the proteins analysed as repetitive. This is supported by the fact that in proteins predicted as repetitive by both sequence and structure-based methods the coverage of tandem repeats is high, but the overlap between both types of predictions is only 29.9%. This further emphasises the complementarity between ordered and disordered RNA-binding regions in tandem repeat RNA-binding proteins.

The differences between sequence and structure-based tandem repeat RNA-binding proteins can also be observed with respect to their biological roles. Both the GO term analysis as well as the identification of Pfam motifs overlapped with trRBDpeps show differences in every subset. Thus, while all the proteins here characterised have in common the RNA-binding function, the sequence-based subset is also related to the interaction with DNA, double-stranded RNA, and mRNA molecules. While the structure-based subset is likely more enriched in functions related with the interaction with other proteins and enzymes, as well as protein-RNA complexes formation. Proteins predicted as repetitive by both types of methods bring together characteristics from both types of repeats.

In conclusion, we have carried out a comprehensive screen for RNA-binding proteins with tandem repeats and found that for the fullest coverage it was necessary to use both sequence and structure based methods for identification of repeats, which can be viewed as complementary in nature. Our results emphasise also the complementary roles of both ordered domains and tandem arrays as well as disordered regions in the RNA-binding process.

## Supporting information

**S1 File.**
(ZIP)

## Acknowledgments

The authors thank Alexander Monzon, Silvio Tosatto and the 2022 REFRACT Hackathon participants for sharing the data about tandem repeat predictors used in this work.

## Author Contributions

**Conceptualization:** Nicolas Palopoli, Alex Bateman.

**Data curation:** Matías Sebastián Carletti, Tadeo Enrique Saldaño, Martín Gonzalez Buitron, Julia Marchetti.

**Formal analysis:** Agustín Ormazábal, Matías Sebastián Carletti, Tadeo Enrique Saldaño, Martín Gonzalez Buitron, Julia Marchetti, Nicolas Palopoli.

**Investigation:** Agustín Ormazábal, Nicolas Palopoli, Alex Bateman.

**Methodology:** Matías Sebastián Carletti, Tadeo Enrique Saldaño, Martín Gonzalez Buitron, Julia Marchetti, Nicolas Palopoli.

**Software:** Agustín Ormazábal, Matías Sebastián Carletti, Tadeo Enrique Saldaño, Martín Gonzalez Buitron, Julia Marchetti.

**Supervision:** Nicolas Palopoli, Alex Bateman.

**Visualization:** Agustín Ormazábal.

**Writing – original draft:** Agustín Ormazábal.

**Writing – review & editing:** Nicolas Palopoli, Alex Bateman.

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
