## [Decision Letter · Decision Letter 0]

16 Aug 2023

Expanding the repertoire of human tandem repeat RNA-binding proteins

PONE-D-23-17094

Dear Dr. Bateman,

We’re pleased to inform you that your manuscript has been judged scientifically suitable for publication and will be formally accepted for publication once it meets all outstanding technical requirements.

Kind regards,

Katherine James, Ph.D.

Academic Editor

PLOS ONE

1. Thank you for stating in your Funding Statement:

“This project has received funding from the European Union’s Horizon 2020 research and innovation staff exchange programme REFRACT under grant agreement No 823886. A.O., M.S.C. and M.G.B. are PhD fellows, J.M. is postdoctoral researcher and N.P. is adjunct researcher from Consejo Nacional de Investigaciones Científicas y Técnicas (CONICET). The work was supported in part by funding from Agencia Nacional de Promoción Científica y Tecnológica (ANPCyT) Grant #PICT-2020-SERIEA-00192 to N.P. The authors of this work are also supported by the core EMBL funding and declare that they have no competing interests. The funders had no role in study design, data collection and analysis, decision to publish, or preparation of the manuscript.”

Please respond by return e-mail so that we can amend your financial disclosure and competing interests on your behalf.

Reviewers' comments:

Reviewer's Responses to Questions

**Comments to the Author**

1. Is the manuscript technically sound, and do the data support the conclusions?

Reviewer #1: Yes

Reviewer #2: Yes

2. Has the statistical analysis been performed appropriately and rigorously? 

Reviewer #1: N/A

Reviewer #2: N/A

3. Have the authors made all data underlying the findings in their manuscript fully available?

Reviewer #1: Yes

Reviewer #2: Yes

4. Is the manuscript presented in an intelligible fashion and written in standard English?

Reviewer #1: Yes

Reviewer #2: Yes

5. Review Comments to the Author

Reviewer #1: The authors have responded appropriately to the prior evaluations of this work offered by myself and by my counterpart in peer review. I especially appreciate the clarified individual case studies offered at the end of the discussion.

No further recommendations or requests for improvement.

Reviewer #2: This paper uses bioinformatic analysis to identify potential tandem repeat RNA binding proteins using both structure prediction and sequence analysis as well as looking at function. Both reviews on the original submission centered around streamlining the paper and intentionally centering on hypothesis and the most interesting points for a biological audience. Although I think it could still be streamlined some more, the authors have made significant modifications based on reviewer comments and I recommend it for publication.

6. PLOS authors have the option to publish the peer review history of their article (what does this mean?). If published, this will include your full peer review and any attached files.

Reviewer #1: No

Reviewer #2: No

---

## [Editor Report · Acceptance letter]

10 Sep 2023

PONE-D-23-17094 

Expanding the repertoire of human tandem repeat RNA-binding proteins 

Dear Dr. Bateman:

I'm pleased to inform you that your manuscript has been deemed suitable for publication in PLOS ONE. Congratulations! Your manuscript is now with our production department. 

Kind regards, 

on behalf of

Dr. Katherine James 

Academic Editor

PLOS ONE